# Bovine Respiratory Disease: Conventional to Culture-Independent Approaches to Studying Antimicrobial Resistance in North America

**DOI:** 10.3390/antibiotics11040487

**Published:** 2022-04-05

**Authors:** Sara Andrés-Lasheras, Murray Jelinski, Rahat Zaheer, Tim A. McAllister

**Affiliations:** 1Lethbridge Research and Development Centre, Agriculture and Agri-Food Canada, Lethbridge, AB T1J 4B1, Canada; sara.andreslasheras@agr.gc.ca (S.A.-L.); rahat.zaheer@agr.gc.ca (R.Z.); 2Western College of Veterinary Medicine, University of Saskatchewan, Saskatoon, SK S7N 5B4, Canada; murray.jelinski@usask.ca

**Keywords:** antimicrobial resistance, bovine respiratory disease, diagnostics, *Histophilus somni*, integrative and conjugative elements, microbiome, *Mannheimia haemolytica*, *Mycoplasma bovis*, *Pasteurella multocida*

## Abstract

Numerous antimicrobial resistance (AMR) surveillance studies have been conducted in North American feedlot cattle to investigate the major bacterial pathogens of the bovine respiratory disease (BRD) complex, specifically: *Mannheimia haemolytica*, *Pasteurella multocida*, *Histophilus somni*, and *Mycoplasma bovis*. While most bacterial isolates recovered from healthy cattle are susceptible to a repertoire of antimicrobials, multidrug resistance is common in isolates recovered from cattle suffering from BRD. Integrative and conjugative elements (ICE) have gained increasing notoriety in BRD-*Pasteurellaceae* as they appear to play a key role in the concentration and dissemination of antimicrobial resistant genes. Likewise, low macrolide susceptibility has been described in feedlot isolates of *M. bovis*. Horizontal gene transfer has also been implicated in the spread of AMR within mycoplasmas, and *in-vitro* experiments have shown that exposure to antimicrobials can generate high levels of resistance in mycoplasmas via a single conjugative event. Consequently, antimicrobial use (AMU) could be accelerating AMR horizontal transfer within all members of the bacterial BRD complex. While metagenomics has been applied to the study of AMR in the microbiota of the respiratory tract, the potential role of the respiratory tract microbiome as an AMR reservoir remains uncertain. Current and prospective molecular tools to survey and characterize AMR need to be adapted as point-of-care technologies to enhance prudent AMU in the beef industry.

## 1. Introduction

The North American beef industry is divided into two main sectors: cow-calf operations and feedlots. Calves are typically born in the spring and raised on pastures where they are seldom administered antimicrobials (Figure 1) [1]. In the fall, the calves are weaned and transported to auction marts where they are commingled, sorted, and sold into feedlots. Feedlots are intensive livestock operations, with calves often being sorted into pens of 200–300 head. Bovine respiratory disease (BRD) is the most common disease affecting newly arrived feedlot cattle. BRD is a multifactorial disease complex in which bacteria, viruses, host, management practices, and environment play an important role (Figure 1) [2]. It has been reported that a high level of comingling at feedlots can predispose cattle to BRD [3]. Although BRD is considered to be a polymicrobial infection, the four most prevalent bacteria associated with this condition are *Mannheimia haemolytica*, *Pasteurella multocida*, *Histophilus somni*, and *Mycoplasma bovis* [4]. With the exception of *M. bovis*, the other three are members of the *Pasteurellaceae*. The epidemiology of BRD is well-known, and hence metaphylaxis therapy is frequently administered to calves upon arrival at the feedlot with the aim of mitigating the incidence and severity of BRD.

BRD accounts for 70–80% and 40–50% of total feedlot cattle morbidities and mortalities, respectively [5]. This costs the North American feedlot cattle industry over $3 billion per year in therapy costs, reduced animal performance, and mortalities [6]. In addition to the monetary losses, BRD is also an animal welfare issue. Despite considerable resources having been invested in the development of pharmaceuticals, vaccines, technologies and management strategies to mitigate and treat BRD, the burden it poses on the North American feedlot industry has remained largely unchanged over the last 45 years [7]. Enhancements in BRD control are challenging due to the complex nature of the disease, diverse cattle management practices, and a lack of economic incentives to reduce BRD through management approaches such as preconditioning [7]. BRD bacterial pathogens have also been shown to become resistant to antimicrobials following use by the cattle industry. Different approaches such as improved diagnostics, probiotics, and more efficacious vaccines continue to be investigated, developed, and validated as alternatives to antimicrobials. However, until these alternatives are shown to be cost-effective, implementable, and exhibit comparable efficacy to current practices, it is likely that antimicrobials will continue to be the mainstay for preventing, treating, and controlling BRD.

Prevention of bacterial diseases through the administration of antimicrobials via the feed or parenterally to a large cohort of cattle or the entire herd is known as metaphylaxis [8]. Metaphylaxis is frequently used at feedlot entry on calves that are at high-risk of developing BRD [9] as it provides therapy to infected animals and prophylaxis (prevention) to uninfected cattle. In Canada, almost 40% of feedlot cattle are deemed to be high-risk, making them 100 times more likely to be administered a macrolide antimicrobial (metaphylaxis) as compared to low-risk calves [9]. Mass medicating for BRD has been questioned because the absolute and relative risk reduction for morbidity and mortality is both modest (relative risk) and highly variable (absolute and relative risks) [10]. There are worldwide efforts towards prudent antimicrobial use (AMU) in human and veterinary medicine, with the goal being to decrease AMR and preserve the efficacy of antimicrobials [11,12,13,14]. Thus, the use of mass medication as a management practice will be increasingly scrutinized by processers, retailers, and consumers [15], emphasizing the need for feasible alternatives to prevent and control BRD.

Bacteria develop AMR via antimicrobial resistance genes (ARG) or nucleotide mutations known as single nucleotide polymorphisms (SNP). *Pasteurellaceae* ARGs can be clustered within integrative and conjugative elements (ICE), which are mobile genetic elements (MGE) integrated in the bacterial host chromosome. ICE are considered the most abundant conjugative elements in prokaryotes, outnumbering conjugative plasmids [16,17]. Multidrug-resistance (MDR) linked to ICE (MDR-ICE) has been described in *Pasteurellaceae* throughout North America, demonstrating their capacity to concentrate ARGs [18,19,20]. Although not linked to ARGs, these mobile elements have also been indirectly associated with the transfer of AMR in mycoplasma species, including *M. bovis* [21,22].

Our knowledge about AMR in BRD bacteria has substantially increased during the last two decades, and culture-independent technologies are being explored to improve BRD diagnosis and assist with prudent AMU. Thus, the objectives of this review are: (I) to provide the current state of AMR in BRD bacteria from active and passive surveillance studies in North America; (II) to discuss the distribution and molecular mechanisms of AMR related to ICE; and (III) to address the study of AMR through culture-independent approaches including novel molecular diagnostic approaches for the rapid detection of AMR. Even though AMR levels in the general feedlot cattle population remain low, integrative and conjugative elements seem to be emerging in BRD *Pasteurellaceae*. Due to the ability they have to concentrate and spread AMR, their presence should be monitored, and their potential impact on antimicrobial treatment failure assessed. Additionally, the study of the ability of MDR-ICE *Pasteurellaceae* to form biofilms and persist in the feedlot environment is of interest. New technologies with rapid diagnostic and chute-side capabilities have the ability to assist with more prudent AMU in the future.

## 2. Surveillance Studies Search Strategy

During December 2021, a search building strategy was followed through the medical subject headings (MeSH) at the PubMed website (https://pubmed.ncbi.nlm.nih.gov/ (accessed on 1 December 2021)), to comprehensibly identify active and passive culture-dependent surveillance studies related to AMR for the principal BRD-bacteria: *M. haemolytica*, *P. multocida*, *H. somni*, and *M. bovis*. The following search terms were used: (“Drug Resistance, Microbial” [MeSH]) AND “Mannheimia haemolytica” [MeSH]; (“Drug Resistance, Microbial” [MeSH]) AND “Pasteurella multocida” [MeSH]; (“Drug Resistance, Microbial” [MeSH]) AND “Histophilus somni” [MeSH]; (“Mycoplasma bovis” [MeSH]) AND “Drug Resistance, Microbial” [MeSH]; (“Anti-Bacterial Agents” [MeSH]) AND “Bovine Respiratory Disease Complex” [MeSH]; and ((“Pasteurellaceae Infections/microbiology” [MeSH]) AND “Cattle” [MeSH]) AND “Microbial Sensitivity Tests/veterinary” [MeSH].

The articles included AMR in BRD clinical cases (passive surveillance) and surveillance studies that assessed AMR in BRD bacteria in the general cattle population (active surveillance) (Table 1 and Appendix A). The focus was Canada and the USA (referred as North America in this manuscript) because of similar management practices between these two countries, which differ from those in Europe and Asia. However, some European studies were included for comparison purposes. For the *Pasteurellaceae*, only those studies that specifically followed Clinical and Laboratory Standards Institute (CLSI) guidelines for antimicrobial susceptibility testing (AST) were included. However, no CLSI guidelines exist for AST in *M. bovis*. The time frame was from January 2000 to December 2021, with a main focus on cattle raised for meat production. Data on dairy isolates were included when AST results were not differentiated between beef and dairy cattle. Only observational studies were included, and experimental studies were excluded. Among AMR reports, only those that possessed minimum metadata, including if the isolates were collected from clinical BRD or healthy cattle, originated from a diagnostic laboratory, stated country of origin and year of isolation, were considered. Additionally, those authors that appeared in ≥3 publications were manually searched in PubMed to ensure that relevant publications were not missed.

## 3. Culture-Dependent Surveillance Studies

A number of culture-dependent methods, such as disk diffusion, broth, and agar dilution, are used for AST of BRD bacteria [42,43]. Using antimicrobial concentration breakpoints, bacteria can be classified as resistant, intermediate, or susceptible (SIR) to specific antimicrobials. There is a trend to replace the AST disk diffusion method [23,27] with the broth microdilution method [31,34,36,41]. Broth dilution has advantages over disk diffusion as the distribution of the minimum inhibitory concentrations (MIC) for each antimicrobial can be reported, allowing for MIC distributions to be monitored in bacterial populations over time and identify changes. Since MIC distributions are publicly available [30,36,37], the data can be reassessed as new antimicrobial breakpoints become available [44]. However, SIR breakpoints are largely undefined for several BRD bacteria and antimicrobial combinations, making comparison of results across AMR studies challenging [43]. To address the scarcity of breakpoints across host, body systems, and bacterial species combinations, the CLSI recently published a document describing valid breakpoint extrapolations by host animal species [45]. The absence of SIR breakpoints for antimicrobials relevant to the cattle industry, such as tylosin (TYL), neomycin (NEO), and chlortetracycline (CTET), has prompted using the presence of ARGs that are known to be associated with higher MIC values to establish “in-house” SIR breakpoints [25].

### 3.1. Pasteurellaceae

Twelve months after tulathromycin (TUL) was approved for use in Canadian cattle (Table 2), a 3-year (2007–2010) TUL resistance study of *M. haemolytica* (*n* = 4548 isolates) was undertaken [24]. Deep nasopharyngeal swabs (DNPS) were used to obtain isolates from calves upon feedlot arrival and exit after a minimum of 60 days on feed (DOF). Sampling involved both healthy cattle and those treated for BRD. The results revealed very low TUL resistance in southern Alberta cattle. Two different publications from the same research group reported AMR patterns in a subset of the same collection (2008–2009) by disk diffusion (*n* = 409) and broth microdilution (*n* = 88), respectively [23,25]. Overall, the level of resistance of *M. haemolytica* to TUL and tilmicosin (TIL) was very low, with resistance to oxytetracycline (OXY) being most common. No differences were observed in AMR between isolates collected from cattle at feedlot entry and exit, nor was there a relationship between the types of antimicrobials administrated and susceptibility profiles. Isolates that presented resistance to at least one antimicrobial were most frequently isolated from morbid cattle (treated with therapeutic antimicrobials) and were identified as *M. haemolytica* serotype A1. A similar study in Alberta sampled beef cattle upon feedlot arrival and later in the feeding period (from 30 to ≥180 DOF) for the evaluation of AMR in 1789 *M. haemolytica* isolates by disk diffusion and 2774 by broth microdilution, with 1574 isolates tested by both methods [26]. Only 6% of the strains were resistant to ≥2 antimicrobials regardless of drug class, with low levels of AMR in *M. haemolytica* isolates from healthy cattle to macrolides, aminoglycosides (AMG), penicillins, phenicols, tetracyclines (TET), cephalosporins (CPH), and fluoroquinolones (FQ).

On the contrary, susceptibilities of *M. haemolytica* recovered at feedlot entry and after 10–14 DOF in North Carolina, revealed an increase in AMR to enrofloxacin (ENRO), florfenicol (FFN), gamithromycin (GAM), TIL, and TUL [27]. Cattle deemed to be at high-risk of developing BRD received TUL upon entry. Similarly, a study of high-risk feedlot cattle in Mississippi and Alabama found increased AMR in *M. haemolytica* isolates from 0 to 21 DOF [28], with high levels of MDR and resistance to TIP, TIL, TUL, FFN, ENRO, and TET identified after seven DOF. In contrast, an Alberta study followed a cohort of steers from spring processing to ≥40 DOF and found that the prevalence of AMR in *M. haemolytica*, *P. multocida*, and *H. somni* to CPH, FQ, TET, β-lactams, and macrolides remained low [29]. Between 2017 and 2019, two large studies conducted in Alberta, Canada, investigated the prevalence of AMR among *M. haemolytica*, *P. multocida*, and *H. somni* in healthy feedlot cattle. The first was a longitudinal study that followed the calves from spring processing to feedlot reprocessing [31]. Metaphylactic use of macrolides at feedlot induction was associated with higher MICs to this drug class in *M. haemolytica*, *P. multocida*, and *H. somni.* The second was a cross-sectional study that sampled beef calves at feedlot entry before the administration of antimicrobials at processing [30]. Overall, AMR prevalence was low, but MICs were higher in isolates from dairy-type versus beef-type feedlot cattle, likely due to higher AMU in dairy as compared to beef calves [1,46].

Amongst the cited studies in this section, AMR in USA isolates from feedlot cattle was generally higher than in Canada. A number of factors could explain this difference: (1) geographical and time variations in AMR; (2) the number of isolates included during the feeding period (Canadian reports assessed a larger number than USA reports), potentially providing a more reliable estimate of AMR levels; (3) USA studies sampled exclusively high-risk cattle that received metaphylaxis at feedlot entry thus, increasing AMR selective pressure; and (4) Canadian studies sampled cattle later in the feeding period (30 to ≥180 DOF) compared to those in the USA (7 to 21 DOF), possibly leading to differences in the detection of transient AMR after feedlot arrival.

Antimicrobial resistance has also been evaluated for bacterial isolates obtained from clinical BRD cases. Three studies reported AMR patterns for *M. haemolytica*, *P. multocida*, and *H. somni*, isolated from clinical BRD cases and mortalities in Canadian and USA commercial feedlots between 1994 and 2011 [32,33,34]. There were trends towards increasing resistance to macrolides, TET, and FQs, while susceptibility to ceftiofur (TIO; 3rd generation cephalosporin, 3GCP) was very high in all BRD bacteria. A study of acute fibrinous pneumonia cases in Canada and the USA found that levels of AMR in *M. haemolytica* and *P. multocida* increased over a 5-year period in cattle exhibiting clinical BRD in the same region [25,35]. The authors reported that 30% and 12.5% of the *M. haemolytica* and *P. multocida* isolates, respectively, were resistant to more than seven antimicrobial classes, including AMG, penicillins, FQ, lincosamides, macrolides, pleuromutilins, and TET. These MIC results were based on bimodal distributions, high MIC values, and the presence of ARG, as official CLSI breakpoints were often unavailable. Interestingly, most of the MDR isolates obtained from the USA possessed ICE, which were absent in isolates from Canada. Higher AMR in isolates from the USA compared to Canada could have been the result of higher AMU due to the larger size and higher animal density of USA feedlots, possibly increasing the incidence of BRD [35].

Sixty different feedlots were enrolled in a study that collected respiratory samples from all types of feedlot cattle (2014–2015; fall and winter calves; yearlings; adults; males, females) from southern Alberta [36]. The authors reported relatively high AMR to macrolides and OXY among *M. haemolytica*, *P. multocida*, and *H. somni*. In agreement with previous studies, only 0.9% of *M. haemolytica* and *P. multocida* isolates were resistant to TIO. In a 2015–2016 study, cattle classified as high-BRD risk were sampled at four different commercial feedlots in southern Alberta [37]. Trans-tracheal samples were taken from morbid cattle during the feeding period and from healthy counterparts. The cattle included in the study were not treated for any infectious disease during the feeding period but received metaphylactic TUL at feedlot arrival. The prevalence of resistant strains was higher among sick than healthy cattle for *M. haemolytica* and *H. somni*, but not for *P. multocida*. Overall, regardless of health status, resistance to OXY was high across the three bacterial species, to macrolides in *M. haemolytica* and *P. multocida*, and to penicillin (PEN) in *H. somni.* Interestingly, higher AMR was found among isolates obtained from morbid cattle, although samples were collected prior to administration of antimicrobials to control BRD.

### 3.2. Mycoplasma Bovis

Inclusion of *M. bovis* in BRD surveillance studies in North America has been inconsistent, likely as a result of its fastidious nature with respect to culturing [47]. Mycoplasmas have small genomes [21] and, unlike BRD-related *Pasteurellaceae*, AMR in *M. bovis* is mediated exclusively via SNPs [48,49]. Official CLSI guidelines and breakpoints for *Mycoplasma* spp. of animal origin are under development for the next edition of CLSI VET06 [50]. As a consequence, substantial variation in AST methodology exists across studies. Agar dilution, agar diffusion, broth microdilution (most frequently used), and Etest all have been used for AST of *M. bovis* [51]. The application of flow cytometry to *M. bovis* AST decreases analysis time and generates results comparable to traditional AST methods [52].

The susceptibilities of 98 *M. bovis* clinical respiratory isolates from USA beef and dairy cattle were investigated from 2002–2003 [38]. *M. bovis* showed sensitivity to ENRO, FFN, and spectinomycin (SPE), intermediate resistance to TET, and resistance to macrolides. *Mycoplasma bovis* was also collected from feedlot cattle purchased at auction over two years (2007–2008) in southern Saskatchewan, Canada [39]. Deep nasopharyngeal swabs were collected at feedlot entry from clinically healthy calves and from cattle exhibiting clinical BRD, as well as swabs from BRD mortalities. *Mycoplasma bovis* isolates from morbidities and mortalities exhibited high MIC_50_ to ERY and TIL, but not to TET, ENRO, FFN, SPE, or TUL. Moreover, the authors reported a positive association between the metaphylactic administration of OXY on entry and the occurrence of chronic pneumonia and polyarthritis syndrome (CPPS)–a condition caused by *M. bovis* [53]. A total of 226 *M. bovis* isolates were obtained from clinical cases (DNPS) and BRD mortalities in a commercial feedlot cattle in southern Alberta [36]. As previously observed, *M. bovis* showed high MIC_50_ to macrolides, intermediate to TET, and low to FQ, SPE, and FFN. *Mycoplasma bovis* isolates from the Animal Health Laboratory at the University of Guelph were investigated for their susceptibilities to different antimicrobials [40]. The authors found overall low FQ, SPE, and TUL MIC_50_ values over three decades. The macrolides TIL and TYL were among those drugs presenting higher MIC_50_ and the effectiveness of OXY, CTET, TIL, and TYL decreased over the study period.

More recently, 156 *M. bovis* isolates collected from the respiratory tract and joints of healthy beef cattle, BRD clinical, and BRD mortalities were tested against the most relevant antimicrobials used to control and treat *M. bovis* infections in Canada [41]. The isolates originated mainly from feedlot cattle in western Canada, but also from Idaho, USA. Overall, MICs tended to increase over time and were higher among isolates obtained from mortalities as compared to healthy cattle. High macrolide-MIC values were identified, whereas most strains were sensitive to ENRO, FFN, and OXY, with modest resistance to CTET. The aforementioned study with beef and dairy type feedlot cattle [30] and a longitudinal study [31] tested a total of 222 and 49 *M. bovis* isolates, respectively. Both studies showed high MIC values to macrolides regardless of cattle type or sampling time.

A common feature found across all *M. bovis* susceptibility studies cited in this review is the low susceptibility of this bacterial species to macrolides. A decrease in the susceptibility to macrolides has been described also in European countries [54,55,56]. These results suggest that macrolides are not the best option to prevent or treat *M. bovis* infections in beef cattle. Among the remaining antimicrobial alternatives, TET should be used with caution because of the possible correlation between its administration and the occurrence of CPPS [39,57].

### 3.3. Occasional BRD Opportunistic Pathogens

*Bibersteinia trehalosi* (*Pasteurellaceae*) is often present in the respiratory tract of domestic and wild sheep [58], bison [59], and occasionally is an opportunistic pathogen in feedlot cattle [60]. A lack of clinical signs, sudden death, and high mortality in spite of aggressive antimicrobial therapy have been associated with sporadic outbreaks of *B. trehalosi* in cattle in both Europe and North America [61,62]. In Europe, *B. trehalosi* is considered to be a relevant member of the BRD complex as it is frequently isolated from cattle [62,63,64]. It has been proposed that the frequency of *B. trehalosi* infections could be on the rise in North America [65], but little is known about its prevalence and role in BRD [66]. It is possible that *B. trehalosi* might have been overlooked or misdiagnosed in the past due to similar colony morphology to *M. haemolytica*, and the tendency to group both bacterial species within the “*Pasteurella haemolytica*” complex [58,63]. Various studies that characterized the respiratory microbiota of feedlot cattle at different production stages and conditions in North America have detected *B. trehalosi* [57,67,68,69,70,71,72,73], possibly reflecting its role as both a commensal and BRD pathogen. Currently, there are no established CLSI guidelines to conduct AST of *B. trehalosi* [43]. While *B. trehalosi* has been found to be consistently sensitive to cephalosporins (MIA-I) [74], few studies have investigated the nature of AMR in this species [59,75,76].

Integrative and conjugative elements have also been identified in *B. trehalosi* (GenBank accession GCA_000521765.1) prompting the question whether it is an AMR reservoir sharing ARGs with other *Pasteurellaceae*. Others have found ARGs previously described in other BRD *Pasteurellaceae*, e.g., *sul2* or *erm* (42), and ARGs related to resistance to phenicols and sulfonamides [65,77,78]. These ARGs are often associated with pCCK13698, a plasmid with insertion sequences (small transposable elements) that are homologous to those in the *Pasteurellaceae* including *M. haemolytica*. Plasmid pCCK13698 was found to be transferable from *B. trehalosi* to *P. multocida*, where it conferred resistance to phenicols and sulfonamides.

*Trueperella pyogenes* is an opportunistic pathogen found in the nasopharynx of cattle, sheep, and pigs [79]. In feedlot cattle, it is mainly associated with liver abscesses [66] where it causes suppurative lesions, but it has also been occasionally implicated as a secondary pathogen of BRD [79]. A number of surveys have investigated AMR in *T. pyogenes*; however, studies with isolates from the bovine respiratory tract are scarce [36,80,81]. These studies suggest that *T. pyogenes* isolated from cattle in North America is frequently resistant to both TETs and macrolides.

## 4. Integrative and Conjugative Elements

Integrative and conjugative elements contain the machinery needed for self excision from the donor cell chromosome, horizontal transmission by conjugation, and integration into the chromosome of the recipient cell [17]. These mobile genetic elements tend to present a modular structure that includes conserved backbone genes coding for basic ICE functions i.e., regulation, excision, conjugation, and recombination, and accessory or cargo genes that can include ARG (Figure 2) [18]. ICE with up to 12 different ARG have been identified in North America in BRD *Pasteurellaceae* [18,19,35,82]. The widespread distribution of ICE that carry ARG (ARG-ICE) is concerning because of their role in the dissemination of AMR through horizontal transfer across bacteria and the potential of MDR-ICE in converting a bacterium from being sensitive to resistant to multiple antimicrobial classes following a single genetic transfer event.

### 4.1. Pasteurellaceae

The first reported *Pasteurellaceae* isolate showing TET-resistance associated with ICE was a *H. somni* isolate from a case of clinical bovine pneumonia in the USA in 2004 (Table 3) [84]. Subsequently, a second ICE-associated TET-resistance was described in *H. somni* in 2018 [85]. Further research identified ICE*Pmu1* (12 ARG) and ICE*Mh1* (5 ARG) in clinical isolates of *P. multocida* and *M. haemolytica*, respectively, from USA cattle [20,82]. Later work found that ICE were present in *M. haemolytica* isolates collected as early as 2002 [86]. Clawson et al. [86] analyzed the genomes of 1,133 *M. haemolytica* isolates from non-clinical and clinical BRD cases in the USA and Canada between 2002 and 2013. Approximately half (607/1133; 53.6%) of all the isolates possessed ICE and of those, 52.7% (320/607) harbored ARG-ICE. The authors described the presence of *M. haemolytica* carrying ARG-ICE among diseased feedlot cattle as early as 2002 and 2008 in the USA and Canada, respectively, and in the general population in the USA in 2013. One of the *M. haemolytica* isolates collected in 2003 possessed ICE containing 12 different ARG (i.e., *aph*A1, *str*B, *str*A, *aad*B, *aad*A15, *sul2*, *floR*, *tet*(H), *bla*_oxa-2_, *erm*(42), *msrE*, and *mphE* genes) encoding resistance to seven different antimicrobial classes [86]. Additionally, *M. haemolytica* with ARG-ICE were recovered from cattle with clinical BRD in the USA [35] and *H. somni*, *P. multocida*, and *M. haemolytica* harboring ARG-ICE were later isolated from southern Alberta feedlot cattle with BRD [18]. Recently, a nucleic acid amplification test (NAAT) called recombinase polymerase assay (RPA) was developed and a collection of nasal isolates from healthy beef cattle entering 10 different southern Alberta commercial feedlots were screened for the presence of ARG-ICE associated with *Pasteurellaceae* [87]. Over half (55%) of the isolates had ARG-ICE, showing that these MGEs are now common in isolates from the general cattle population in Canada.

### 4.2. Mycoplasma Bovis

Mycoplasma ICE can be transferred via conjugation within the same and between different mycoplasma species. Transfer of virtually any region within the mycoplasma chromosome by massive homologous recombination (HR; DNA fragments of up to 90 kb) can occur [22]. Mycoplasmas may contain multiple copies of ICE within the same genome [22], and although also modular, their structure differs from the ICE in *Pasteurellaceae* (Figure 2). One of the main differences between *Pasteurellaceae* and mycoplasma ICE is the absence of ARG-ICE linkage and the reliance on SNPs to confer AMR within mycoplasmas. For example, macrolide resistance is mediated via specific mutations in the 16S and 23S rRNA genes, whereas TET and FQ resistance is mediated by SNPs in 16S and 23S rRNA, and *gyrA*, *gyrB*, and *parC* genes, respectively (Table 4) [51].

*Mycoplasma agalactiae* has been used as a model for *in silico* and *in vitro* analysis of conjugation with *M. bovis* [88]. *Mycoplasma bovis* appears to harbor ICE more frequently than other mycoplasmas within their *Hominis* group [88]. Interestingly, the transfer and replacement of genes related to quinolone-resistance (i.e., *parE* and *parC*) from *M. bovis* to *M. agalactiae* can occur via conjugation [22]. Furthermore, Faucher et al. (2019) found that *in vitro* exposure of *M. agalactiae* to fluoroquinolones increased resistance as a result of conjugative transfer of several resistant alleles [89]. The increasing resistance to macrolides in mycoplasma species of human and veterinary origin is of concern [41,51,56,90,91]. Macrolide-resistant *M. bovis* have been isolated from beef cattle at feedlot entry prior to the administration of antimicrobials [30,41]. Mycoplasmas are known to have a high rate of nucleotide substitution [92], and one could hypothesize that a high substitution rate combined with ICE-mediated homologous recombination could be responsible for their high resistance to macrolides. To date, no studies have been conducted with regard to the epidemiology of ICE in *M. bovis* isolated from feedlot cattle.

## 5. Culture-Independent Surveillance Studies

Only a few studies have used metagenomic approaches to investigate AMR in the microbiota of the respiratory tract of feedlot cattle in North America, using either quantitative PCR (qPCR) or metagenomic DNA sequencing (Table 5) [29,57,68,69]. The relative abundance of AMR genes following OXY and TUL administration to high-risk beef cattle at feedlot entry was evaluated by Holman et al. (2018) [68] using nasopharyngeal samples taken from commercial cattle at feedlot entry and after ≥60 DOF. The results showed that the prevalence of *tet*(H) significantly increased between both sampling points in the OXY treated group, whereas the proportion of *sul1* decreased. In contrast, the relative abundance of *sul2* and *tet*(W) was similar between the two sampling points. Another study investigated the effects of the same drugs on the nasopharynx and gut microbiota of beef cattle after feedlot placement [57]. Both antimicrobials increased the relative abundance of *erm*(X), *sul2*, and *tet*(M), whereas OXY increased *tet*(H) and *tet*(W) as well. Noteworthy, the effects that OXY had on the relative abundance of antimicrobial determinants was observed after 14 DOF, whereas it was after 36 DOF for cattle treated with TUL. Interestingly, a higher relative abundance of *sul2*, *tet*(H) *tet*(M), and *tet*(W) was observed at 36 DOF among all treatments including the control group (i.e., no antimicrobials). The detection of comparable antimicrobial determinant levels in the control and treated cattle could have been a consequence of environmental transmission of resistant bacteria/determinants across individuals within different experimental groups.

Bronchoalveolar lavage samples (BAL) were taken from feedlot cattle mortalities for the study of the lower respiratory tract microbiome and resistome [69]. The metagenomic DNA from 15 BRD-related mortalities and three non-BRD mortalities were sequenced. Not surprisingly, the most abundant ARGs found were related to antimicrobials used in the feedlot industry i.e., TET, TIO, FFN, macrolides, trimethoprim/sulfamethoxazole, and sulphadoxine. Strikingly, the *bla*_ROB-1_, a β-lactamase-encoding gene, was two to ten times more abundant than any other AMR gene. In Canada, β-lactams are mainly used only for therapeutic purposes in feedlot cattle, thus having a limited usage compared to metaphylactically used TET or macrolides [9]. In the scenario of theses findings, the authors questioned whether the volume of AMU alone can be used to predict ARG prevalence within bacterial populations isolated from livestock. In support of these findings, resistance to TIO was extremely low in BRD bacteria isolated in both passive and active surveillance studies. However, how bacteria respond to antimicrobial exposure may be species-dependant. Ceftiofur is more frequently used in dairy than in beef cattle [9,46]. Yet, BRD bacteria from dairy origin did not show high TIO-resistance [30,76], whereas dairy enteric bacteria such as *Salmonella* spp. did [93]. This suggests that AMU may not equally affect the development of AMR in different bacterial species. Considering that cephalosporins are widely distributed in most body fluids and tissues [94], a difference in TIO-resistance between enteric and respiratory bacteria does not seem to be a result of differences in tissue associated antimicrobial concentration. Klima et al. [69] also investigated the presence of ICE and their relationship to AMR determinants. Integrative and conjugative elements were most prevalent in *M. haemolytica* and *P. multocida* but were also identified in *Bacteroides ovatus* (human pathogen) and *Bacteroides xylanisolvens* (human commensal). Additionally, AMR genes not previously described in *Pasteurellaceae* were identified among these samples, raising the possibility that non-BRD bacteria may act as a reservoir of AMR in the bovine respiratory tract.

A cohort of steers was used to evaluate the evolution of AMR from spring processing to ≥40 DOF [29]. Deep nasopharyngeal swabs were taken from cattle housed in three different ranches and feedlots. Animals were administered antimicrobials and vaccines as per industry standards. Interestingly, an increasing trend for *sul2* and *tet*(H) was observed after feedlot placement in the herd that had received in-feed CTET, suggesting that metaphylaxis selected for AMR. However, a correlation between AMU and AMR is not always clear among BRD bacteria [26,95]. Rather, time spent in the feedlot environment appears to have a greater effect on AMR than antimicrobial administration *per se.* This suggests that the feedlot environment could serve as a source of AMR bacteria and/or antimicrobial residues that could exert selective pressures for AMR in environmental bacteria [96,97]. Recently, the persistence of ARGs was evaluated in soil samples collected from a semi-intensive beef cattle backgrounding facility two years after it was decommissioned [98]. The operation was divided into feeding and grazing areas separated by a fence. The ARGs coding for resistance to macrolides, sulfonamides, and TETs were still present after two years, with higher levels detected near the feed bunks and water troughs compared to the pasture zone. The authors attributed this observation to fecal deposition promoting the persistence of AMR bacteria at these sites.

## 6. Novel Approaches for the Rapid Detection of Antimicrobial-Resistant BRD Bacteria

The process from respiratory specimen collection to bacterial antimicrobial susceptibility testing (AST) of individual bacterial isolates can be time consuming. A number of different technologies that have been assessed for their ability or have the potential to detect AMR-BRD bacteria in a timely manner are discussed in this section (Table 6). As an alternative, genome sequencing of individual bacterial isolates has the potential to provide AMR data and overcome the intrinsic variability of traditional AST, while providing relevant information on virulence, metabolic profile, and environmental fitness. Metagenomic, culture-independent strategies skip bacterial isolation and have the potential to reduce time to results, providing AMR genomic information from both pathogenic and non-pathogenic bacteria that could serve as a reservoir of ARG. Approaches that require initial bacterial isolation from an animal specimen also have the potential to shorten time to results if they rely on non-culture AST profiling such as MALDI-TOF (matrix assisted laser desorption ionization-time of flight).

Recombinase polymerase amplification (RPA) is a simple, sensitive, and rapid technology based on isothermal DNA-amplification with a multiplexing option [104]. It is less sensitive to the presence of amplification inhibitors than conventional PCR. The amplicons generated by RPA can be evaluated in real-time using fluorescent probes, visualized by gel electrophoresis, or by lateral flow strips [105]. Compared to qPCR, RPA is less expensive and faster with an execution time of <30 min [105]. All these characteristics make RPA suitable for point-of-care diagnostics. Recently, different multiplex and real-time RPA assays were developed for the detection of the major BRD bacteria (i.e., *M. haemolytica* serotypes A1 and A6, *P. multocida*, *H. somni*, and *M. bovis*) and ARGs related to *Pasteurellaceae* ICE in metagenomic DNA collected from DNPS [87]. Although RPA is a rapid culture-independent method, it still requires extraction of DNA from DNPS prior to its execution.

Another detection technology is loop-mediated isothermal amplification (LAMP), which enables DNA amplification at a constant temperature, resulting in high sensitivity, specificity, and a short turnaround time [100]. LAMP is also stable in the presence of PCR inhibitors such as blood and can be applied without formal DNA extraction [100]. Its high specificity relies on a high number of primers (4 to 6) for the amplification of a single gene target, making primer design challenging, and limiting multiplexing. Nevertheless, the results of single tests can be visualized based on chromophore formation or turbidity, eliminating the need for specialized equipment [100,106,107]. Multiplex LAMP assays can also be designed to enable visualization of results through a lateral flow biosensor, although the analysis time for this approach is more than 1 h [108]. LAMP has been used for the detection of *P. multocida*, *M. haemolytica*, *H. somni*, and *M. bovis* from bovine and swine respiratory samples, as well as for the detection of AMR (ARGs and mutations) in non-BRD bacterial species (e.g., *Mycoplasma pneumoniae* or *Staphylococcus* spp.) [106,107,109,110,111]. Moreover, Pascual-Garrigos et al. (2021) [107] reported the on-farm use of LAMP on crude bovine respiratory samples for the simultaneous detection of BRD *Pasteurellaceae*, demonstrating the utility of LAMP for field applications.

Molecular approaches used in diagnostics present their own advantages and disadvantages (Table 4). DNA isothermal amplification technologies offer rapid and accurate detection of pathogens and ARGs. However, the detection of DNA from non-viable bacterial cells is always a risk, resulting in a possible over-estimation of the presence of BRD bacterial agents and/or ARGs. This over-estimation could be overcome by the use of propidium monoazide (PMA) and sodium deoxycholate (SDO) coupled with PCR, LAMP, or RPA [112,113,114]. Propidium monoazide is a DNA-intercalating dye that prevents interference of DNA from dead cells, whereas SDO is a detergent with the ability to disrupt the outer membrane of injured or dead cells, enhancing the effect of PMA. Therefore, the combination of both substances enables the detection of only viable bacterial cells. Another disadvantage of nucleic acid amplification-methods is their dependency on continuous target updates in order to include emerging resistance genes and mutations which could limit assay scope.

Metagenomic applications have shown promising results in the rapid diagnostics field. MinION (Oxford Nanopore Technologies) is a third generation sequencer that produces long reads that can be analyzed in real-time; is highly portable; has a relatively low cost and can be interfaced with a desktop or laptop computer for data analysis [102]. Nanopore sequencing metagenomics (MinION device) has been used to identify human respiratory bacterial pathogens and ARGs in the lower respiratory tract within 6–8 h [115]. This methodology demonstrated high sensitivity and specificity, could be coupled to procedures that eliminated 99.99% of host nucleic acids, and promoted an early and targeted therapy that supported antimicrobial stewardship. Nanopore sequencing technology has the potential to simultaneously detect both pathogens and ARGs. However, the detection of ARGs that are not associated with pathogenic bacteria [115] or that are at very low concentration as in the respiratory tract of healthy cattle [69], remains a challenge. A possible solution to these disadvantages is the use of genome reduction methods. These methods focus metagenomic sequencing on a reduced number of genes, resulting in higher coverage and sensitivity. An example of a genome reduction technique is the use of custom bait (probe) sequences: these probes bind with the targeted DNA regions of metagenomic DNA. Once the hybridized DNA-probe molecules are captured, they can be PCR-amplified, and sequenced [116]. Battery powered equipment has been used for *in-situ* sample preparation, i.e., DNA extraction and library preparation, for MinION metagenomic sequencing under field conditions [102]. However, long-read point-of-care sample preparation procedures still require optimization for pathogen and ARG detection, particularly with regard to reducing interference from host nucleic acids [102]. A combination of DNA host depletion of the sample coupled with nanopore adaptive sequencing could improve the sequencing efficiency of low abundance targets [117]. Nanopore has been used to sequence the individual genomes of. *M. haemolytica* and *M. bovis* [118,119], but it has not been tested for diagnostic purposes in bovine metagenomic respiratory samples. Long-reads metagenomic sequencing could be highly useful for the identification of *M. bovis* gene mutations that result in AMR because, unlike ARGs, they cannot be detected by a simple PCR without implementing post-amplification sequencing-based procedures.

While third generation sequencing evolves towards becoming a routine procedure for the phenotypic prediction of AMR in metagenomic samples, there are current technologies that are already being used in diagnostic laboratories for the rapid detection of AMR in individual bacterial isolates. An excellent example of this is MALDI-TOF, as it provides AST results faster than conventional methods that require bacteria to be cultured. This technology has been used for the detection of ARGs (e.g., *vanA*, 96.7% sensitivity and 98.1% specificity), activity quantification of antibiotic-inactivating enzymes (e.g., carbapenemases, 92.4% sensitivity and 97.4% specificity), and measurement of bacterial growth exposed to different antimicrobials [120,121,122]. For example, the MALDI Biotyper antibiotic susceptibility test successfully identified TET resistance in 98% of *P. multocida* isolates tested (*n* = 100) within 3 h after isolation [123].

## 7. Further Considerations

Based on the results obtained in previous AMR surveillance studies, *Pasteurellaceae* AMR in beef cattle is low at feedlot entry [23,26,30] and increases over the feeding period [25,31,32,33,34,35,36] likely as a consequence of increased exposure to antimicrobials when calves are transferred from cow-calf operations to feedlots (Figure 3).

### 7.1. Understanding the Genomic Epidemiology of ICE

The emergence of AMR among BRD *Pasteurellaceae* appears to be linked to the emergence of ARG-ICE within these bacterial populations. In the early 2000s, passive surveillance studies identified increasing AMR trends that were likely associated with the emergence of ARG-ICE within *Pasteurellaceae* [32,33,34]. Supporting this hypothesis, other studies suggested that the propagation of ICE in *Pasteurellaceae* is a relatively recent evolutionary event based on the level of conservation across non-AMR ICE genes [83]. Nevertheless, the evolutionary history of ICE in *Pasteurellaceae* has not been formally investigated. Such studies could reveal how long MDR-ICE have been in *Pasteurellaceae* isolated from beef cattle and when they acquired ARG. More historical *Pasteurellaceae* collections should be screened for the presence of ICE linked to ARGs to investigate how prevalent ICE were in the past. Moreover, *Pasteurellaceae* isolates containing ICE recovered from feedlot BRD mortalities should be compared with counterparts obtained from the general feedlot cattle population and their phylogenetic association determined.

Klima et al. [25] and Noyes et al. [37] detected very low AMR in *M. haemolytica* at feedlot entry. However, the sampling periods of both studies took place around the same period (2007–2010) when ARG-ICE were first detected in Canadian *M. haemolytica* isolates (2008) [86]. Recently, MDR *Pasteurellaceae* isolates presenting up to 13 ARGs were isolated from feedlot cattle in western Canada and, although at low prevalence, they were more common in feedlot dairy than beef calves [30]. ARG-ICE appear to be more prevalent among *Pasteurellaceae* isolates obtained from BRD clinical cases than healthy cattle [18,30,35,86], further supporting the possibility that AMU during the feeding period, specially metaphylaxis, could be selecting for ARG-ICE isolates. Because of the risk that these strains could pose to the feedlot industry, the epidemiology of ICE in *Pasteurellaceae* should be evaluated in the general feedlot cattle population as well as their possible emergence and impact on BRD treatment failure and mortality (Figure 3).

The interdisciplinary approach of combining genomics and systematically collected epidemiological data should be the cornerstone of future AMR studies, and for informing the prudent use of antimicrobials. ICE described in *Pasteurellaceae* can vary substantially in structure, and ARG can also be associated with other MGE or the bacterial chromosome [83]. Interestingly, higher numbers of ARGs (up to 12) were found in ICE from *M. haemolytica* isolated from BRD clinical cases than in those obtained from healthy cattle (up to five) [82,86]. Genes encoding resistance to macrolides and tetracyclines can frequently coexist within the same ICE, along with florfenicol, aminoglycosides, β-lactams, or sulfonamides ARG [18,20,82]. The higher number of ARG within ICE detected in *Pasteurellaceae* isolated from clinical BRD cases could be a consequence of co-selection for other ARGs. Newer statistical approaches, such as additive Bayesian networks (ABN), could assist in elucidating the systematic associations amongst phenotypic antimicrobial resistance or ARG in *Pasteurellaceae* from cattle [124]. Coupling ABN with a risk assessment analysis [125] could help assess which antimicrobials are more frequently linked to the selection of resistance to multiple antimicrobials (co-resistance) in feedlot cattle *Pasteurellaceae*.

*In vitro* ICE can be transferred via conjugation between *Pasteurellaceae* bacteria and to other non-*Pasteurellaceae* species, especially γ-proteobacterial pathogens of clinical importance [35,126]. However, the extent to which the respiratory microbiome serves as a reservoir of ARGs for *Pasteurellaceae* and the magnitude of MGEs exchange at the population level within the bovine respiratory tract is unknown (Figure 3). Other bacterial species commonly found in the respiratory tract, such as *Moraxella*, may conjugate with *Pasteurellaceae* [69,127,128]. Transcriptomics could assist in the evaluation of ICE transfer within bacterial communities based on the expression of key ICE genes that mediate excision, transfer, and chromosomal re-integration. There is no conclusive evidence whether antimicrobials directly regulate the efficiency of HGT or just pose a selection force that modulates population dynamics after HGT has occurred [129]. The possible impact that antimicrobials may have on *Pasteurellaceae* ICE conjugation also deserves further investigation (Figure 3).

Mobile genetic elements, including ICE, are able to acquire new traits such as ARGs [83,128]. Noteworthy, ICE with no ARGs have been described in *M. haemolytica* obtained from healthy feedlot cattle in Canada and the USA [19,86]. Whether the presence of “empty” ICE in *Pasteurellaceae* represents an opportunity to acquire ARGs, and contribute to AMR, remains to be elucidated. However, superinfection exclusion systems have been described in other types of ICE e.g., SXT-R391 ICE family or ICE*Bs*1 [17]. These systems prevent the entry of new ICE into a bacterial cell if it already contains a closely related ICE. Potential gene homologs to key genes found in other exclusion mechanisms have been described in BRD *Pasteurellaceae* i.e., *traG* and *eex* [19], but it is unknown if these are functional. However, the fact that only one ICE per cell has been described in BRD *Pasteurellaceae* [18] suggests that this exclusion mechanisms could be functional. The presence of functional exclusion systems in *Pasteurellaceae* ICE could be limiting the horizontal spread of MDR-ICE rendering vertical/clonal expansion their main method of dissemination and may explain why their prevalence in the general feedlot cattle population is lower than in cattle with BRD. However, ICE clonal expansion could pose a risk of MDR-ICE dissemination within sick feedlot pens. Calves diagnosed with an infectious disease are often temporarily relocated to a ‘sick’ pen where they mingle with other sick calves, often treated with multiple antimicrobials. With high AMU, sick pens are more likely to concentrate AMR bacteria that have a high potential to be transferred to other immunocompromised individuals within the pen. In mycoplasmas, more than one ICE copy can be found within a cell genome [22], raising the question whether mycoplasmas possess ICE exclusion systems or if they undergo homologous recombination with no obvious control systems.

### 7.2. Strategies to Mitigate ICE

Strategies designed to limit the spread of ARG-conjugative plasmids could limit the spread of ICE. Among these, the CRISPR/Cas system has shown promise in reducing the acquisition and transmission of AMR via MGEs in *Staphylococcus epidermidis*, *Streptococcus pyogenes*, and *Escherichia coli* [130]. This approach could be beneficial as it could be designed to specifically target *Pasteurellaceae* ICE by either blocking their spread (blocking conjugation) or targeting their ARGs (Figure 4). With this approach, CRISPR/Cas could be designed to specifically recognize and target conserved ICE DNA regions that would trigger a bacterial “immune response” leading to ICE cleavage as new conjugation events take place. Targeting specific ARG by CRISPR/Cas would lead to either the death of those bacterial cells containing the ARG of interest or the loss of the plasmid harboring them. Molecular approaches can also be used for ICE epidemiological surveillance purposes. Beker et al. (2018) [83] developed a multiplex PCR to detect conserved ICE genes within the *Pasteurellaceae* that could assist in monitoring their epidemiology with the purpose of mitigating ARG-ICE spread and guide prudent AMU.

### 7.3. Defining the Fitness of BRD Pathogens in the Environment

Feedlots receive a constant flow of new animals during the feeding season, and pens are cleaned but not sterilized, which could allow AMR bacteria to persist in this environment. *Mannheimia haemolytica*, *P. multocida*, and *H. somni* have been detected on house flies captured on commercial feedlots and on grass, drinking water, and bedding in sheep farms [131,132], indicating they can survive in the environment. Considering that feedlot cattle can carry ICE-MDR *Pasteurellaceae* bacteria upon entry, environmental studies would be of interest to determine whether the feedlot environment serves as a significant source of AMR for calves, specifically, if BRD bacteria containing ARG-ICE persist overtime in feedlot sick pens (Figure 3).

### 7.4. Role of Biofilms in AMR within BRD Pathogens

Antimicrobial resistance genes are not the only factor impacting the occurrence of BRD in cattle as viruses, immune status, and the formation of biofilms can all influence the efficacy of antimicrobial therapy. Antimicrobial susceptibilities are typically assessed using planktonic cells, but it is known that the majority of infections are associated with bacteria within biofilms [133]. Biofilms are structured microbial communities that are embedded in an extracellular polymeric substance (EPS) or matrix and attached to a surface (Figure 5). Biofilms present phenotypic characteristics that differentiates them from free-living or planktonic bacterial cells such as exhibiting higher levels of antimicrobial resistance [134]. The Calgary Biofilm Device (CBD) was specifically designed to test the antimicrobial susceptibilities of biofilms [135]. The susceptibilities of *M. haemolytica* and *P. multocida* isolates obtained from bovine pneumonia were CBD-tested showing no AMR differences when compared with planktonic susceptibility testing [136]. However, another study found that *M. haemolytica* presented higher resistance to FFN, gentamycin, and TUL in biofilms as compared to planktonic cells [137]. *Trueperella pyogenes* was found to be highly resistant to PEN, cloxacillin, streptomycin, TIO, TET, AMP, and OXY in biofilms, as compared to planktonic bacteria [136]. In contrast, *Mycoplasma bovis* biofilms did not exhibit increased resistance to FQ and TET compared to their planktonic counterparts [138]. Biofilms are complex communities that contain different bacterial species within their matrix that evolve from initial colonies to form complex single- or multi-species communities [134]. Contradictory biofilm susceptibility results within the same bacterial species or failing to detect an association between biofilm and higher AMR, may be influenced by the bacterial species investigated and/ or the growth stage of the biofilm at the time of antimicrobial exposure.

## 8. Conclusions

Continuous surveillance of AMR in BRD bacteria provides veterinarians and producers with geographically relevant and contemporaneous AMR to support the prudent use of antimicrobials in the feedlot industry. Noteworthy, the Canadian beef industry in partnership with CIPARS (Canadian Integrated Program for Antimicrobial Resistance Surveillance) initiated a national feedlot AMU/AMR surveillance program that targets selected respiratory and enteric bacterial pathogens [139]. The interdisciplinary approach of combining genomics and sound epidemiology should be the cornerstone for future AMR studies. Since ICEs are more commonly reported in BRD bacteria, their presence in the general feedlot cattle population (i.e., active surveillance) should be monitored and their potential impact on antimicrobial treatment failure assessed. Additionally, environmental studies would be of interest to determine whether BRD bacteria with ICE persist in the feedlot environment; new technologies with rapid diagnostic capabilities could assist with these purposes.

## Figures and Tables

**Figure 1 antibiotics-11-00487-f001:**
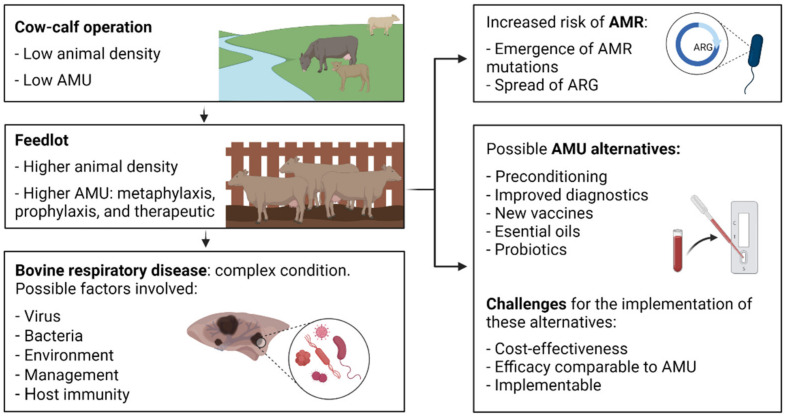
Possible predisposing factors and common management practices to control bovine respiratory disease. AMU, antimicrobial use; ARG, antimicrobial resistance genes; BRD, bovine respiratory disease. Created with BioRender.com.

**Figure 2 antibiotics-11-00487-f002:**
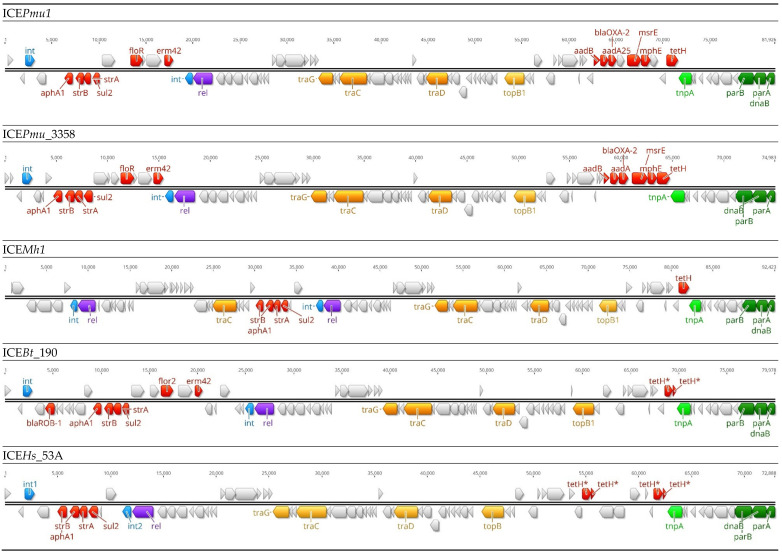
Schematic representation of integrative and conjugative elements from different BRD bacterial species. Genes are represented as arrows, with the arrowhead indicating the direction of transcription. ICE size can be read from the base pair scale on each figure (grey numbers). *Pasteurellaceae* ICE (A–E): red, antimicrobial resistance genes (ARG); blue, integrase (*int1* or *int2*); purple, relaxase (*rel1* or *rel2*); dark green, DNA replication genes (*parA*, *parB*, *dnaB*, and *topB*); light green, transposase (*tnpA*); orange, conjugative transfer (*traB*, *traC*, *traD*, and *traG*); grey, other CDS. ICE-core genes and ARG were annotated as per [18,83] (Geneious v.10.2.6). ICE*Bt*_190, ICE from *B. trehalosi* USDA 190 (CP006956.1); ICE*Hs*_53A, ICE from *H. somni* 53A (adapted from Ref [18]; PRJNA605035); ICE*Mh1*, ICE from *M. haemolytica* 42548 (CP005383.1); ICE*Pmu1*, ICE from *P. multocida* 36950 (CP003022.1); ICE*Pmu*_3358, ICE from *P. multocida* 3358 (CP029712.1). *Mycoplasma agalactiae* 5632 ICEA-I (F) (CT030003.1): light green, transposase; light orange, CDS candidates for conjugative channel (CDS5-19); dark orange, CDS14 has a key role in ICE and chromosome transfer [21]. * Partial sequence.

**Figure 3 antibiotics-11-00487-f003:**
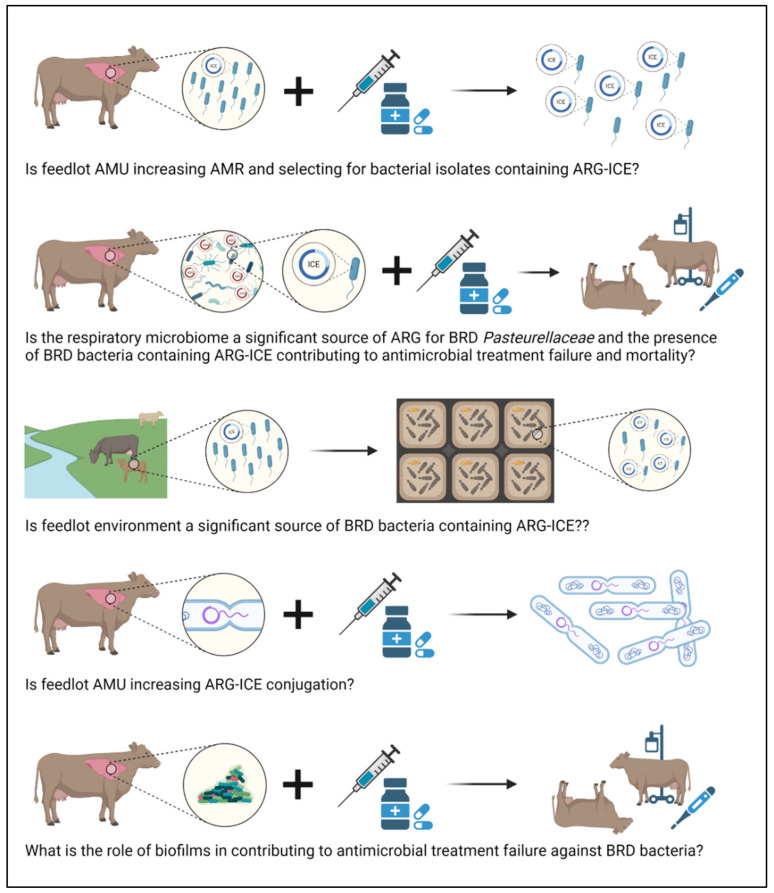
Main knowledge gaps related to antimicrobial resistance in bovine respiratory disease bacteria. ARG, antimicrobial resistance gene; ARG-ICE, integrative and conjugative elements linked to antimicrobial resistance genes; AMU, antimicrobial use; BRD, bovine respiratory disease. Created with BioRender.com.

**Figure 4 antibiotics-11-00487-f004:**
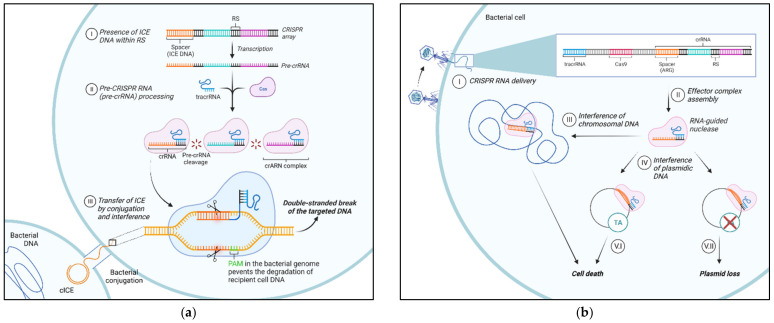
Possible clustered regularly interspaced short palindromic repeats (CRISPR)-based approaches to limit the spread of antimicrobial resistance. (**a**) Conjugation blockage: (I) integrative and conjugative element (ICE) DNA fragments (spacers) are present in the bacterial genome in between the repeated CRISPR sequences (RS) as a consequence of spacer acquisition from previous ICE exposure; (II) trans-activating CRISPR RNA (tracrRNAs) recognize CRISPR RNA (crRNA) sequences (pre-crRNA); one or more Cas proteins process the long pre-crRNA into individual RNA, called guide ARN or crARN, which forms a complex with one or more Cas proteins; (III) when the bacterium receives the same ICE by conjugation, the crRNA complex recognizes protospacer adjacent motif (PAM) motifs leading to target invading DNA recognition and Cas-mediated cleavage (**b**) Targeting antimicrobial resistance genes: (I) The RNA-guided nuclease (RGN) construct is delivered inside the targeted bacteria by a bacteriophage; (II) CRISPR locus is transcribed and processed into crRNAs, that together with tracrRNA and Cas proteins, form the RGN; (III) when the targeted DNA sequence is present in the bacterial chromosomal DNA, the RGN activity is cytotoxic; (IV) when the targeted DNA sequence is present in the bacterial plasmid DNA, there are two possible outcomes depending on the presence/absence of toxin-antitoxin (TA) systems; (V.I) TA system is present: the RGN activity is cytotoxic; (V.II) TA system is absent: the RGN activity leads to plasmid loss and resensitization the bacterial cell. cICE, circular ICE. Created with BioRender.com.

**Figure 5 antibiotics-11-00487-f005:**
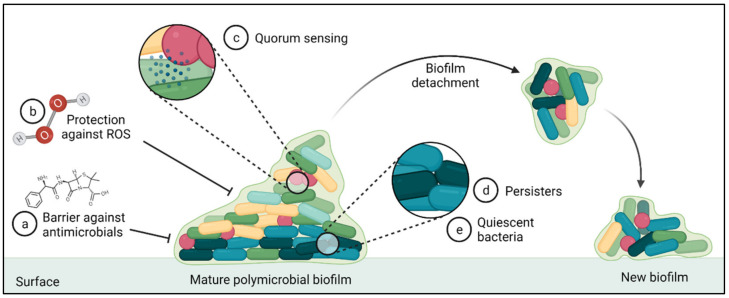
Different mechanisms that contribute to the resilience of biofilms and higher antimicrobial resistance. (a) Extracellular polymeric substance (EPS) that acts as a barrier to the diffusion of antimicrobials and other substances. (b) Up-regulation of the protective mechanisms against reactive oxygen species (ROS). (c) Bacterial signaling with the ability to activate virulence factors that also contributes to biofilm formation and detachment, or resistance against ROS amongst others. (d) Presence of highly-antimicrobial resistance bacterial sub-populations called persisters. (e) Bacteria present in the lower layers of a biofilm that present differential gene/protein expression and slower rates of growth that reduces effectiveness of antimicrobials. Source: [134]. Created with BioRender.com.

**Table 1 antibiotics-11-00487-t001:** Summary of the North American culture-dependent surveillance studies of bacteria associated with bovine respiratory disease in beef cattle discussed in this review (see Section 3).

Geographical Area	Sampling Year (s)	Targeted Bacteria	Authors
*Pasteurellaceae* AMR active surveillance studies			
Southern Alberta, Canada	2008–2009	Mh	Klima et al. 2011 [23]
Southern Alberta, Canada	2007–2010	Mh	Alexander et al., 2013 [24]
Southern Alberta, Canada	2008–2009	Mh	Klima et al., 2014 [25]
Southern and central Alberta, Canada	2007–2010	Mh	Noyes et al., 2015 [26]
Central Georgia, USA	2016	Mh	Snyder et al., 2017 [27]
Mississippi, USA	2016	Mh, Pm, Hs	Woolums et al., 2018 [28]
Southern Alberta, Canada	2016	Mh, Pm, Hs	Guo et al., 2020 [29]
Alberta, Canada	2017–2019	Mh, Pm, Hs	Andres-Lasheras et al., 2021 [30]
Alberta, Canada	2017–2017	Mh, Pm, Hs	Nobrega et al., 2021 [31]
*Pasteurellaceae* AMR passive surveillance studies			
Oklahoma Animal Disease Diagnostic Laboratory, USA	1994–2002	Mh, Pm, Hs	Welsh et al., 2004 [32]
Veterinary diagnostic laboratories across USA and Canada	2000–2009	Mh, Pm, Hs	Portis et al., 2012 [33]
Kansas State Veterinary Diagnostic Laboratory, USA	2009–2011	Mh	Lubbers and Hanzlicek. 2013 [34]
Alberta (Canada), Texas, and Nebraska (USA)	NS	Mh, Pm, Hs	Klima et al., 2014 [35]
Southern Alberta, Canada	2014–2015	Mh, Pm, Hs	Anholt et al., 2017 [36]
Southern Alberta, Canada	2015–2016	Mh, Pm, Hs	Timsit et al., 2017 [37]
*Mycoplasma bovis* AMR active and passive surveillance studies			
Different regions across USA	2002–2003	Mb	Rosenbusch et al., 2005 [38]
Saskatchewan, Canada	2007–2008	Mb	Hendrick et al., 2013 [39]
Southern Alberta, Canada	2015–2016	Mb	Anholt et al., 2017 [36]
Animal Health Laboratory at the University of Guelph, Canada	1978–2019	Mb	Cai et al., 2019 [40]
Western Canada and Idaho, USA	2006–2018	Mb	Jelinski et al., 2020 [41]
Alberta, Canada	2017–2019	Mb	Andres-Lasheras et al., 2021 [30]
Alberta, Canada	2017	Mb	Nobrega et al., 2021 [31]

Hs, *Histophilus somni*; Mb, *Mycoplasma bovis*; Mh, *Mannheimia haemolytica*; NS, not specified; Pm, *Pasteurella multocida*. Active surveillance: the study of AMR in the general cattle population; Passive surveillance: the study of AMR in BRD clinical cases.

**Table 2 antibiotics-11-00487-t002:** Year of license approval for antimicrobials used in North American feedlot cattle for the prevention and treatment of bovine respiratory disease (BRD).

Antimicrobial	Registration	Common Use (s)
Tildipirosin	2012	Treatment of BRD
Gamithromycin	2010	Treatment of BRD
Tulathromycin	2005	Prevention/treatment of BRD
Enrofloxacin	2004	Treatment of BRD
Danofloxacin	2002	Treatment of BRD
Tylosin	1997	Prevention/treatment of liver abscess
Florfenicol	1996	Treatment of BRD and footrot
Spectinomycin	1996	Treatment of BRD
Chlortetracycline	1995	Prevention of footrot and BRD; prevention/treatment of liver abscess; treatment of enteritis
Ceftiofur	1994	Treatment of BRD
Oxytetracycline	1994	Prevention/treatment of liver abscess; prevention/treatment of BRD; prevention of bloat
Trimethoprim	1994	Treatment of BRD
Tilmicosin	1992	Prevention/treatment of BRD

BRD, bovine respiratory disease.

**Table 3 antibiotics-11-00487-t003:** Summary of the North American studies that detected integrative and conjugative elements in bovine respiratory disease bacteria isolated from feedlot cattle.

Geographical Area	Sampling Year (s)	Bacterial Species	Number of ARGs Reported Linked to ICE	Authors
USA	NS	Hs	1	Mohd-Zain et al., 2004 [84]
Nebraska, USA	2005	Pm	12	Michael et al., 2012 [82]
Texas and Nebraska, USA	NS	Mh	Up to 14	Klima et al., 2014 [35]
Pennsylvania, USA	2007	Mh	5	Eidam et al., 2015 [20]
Canada and USA	2002–2013	Mh	Up to 12	Clawson et al., 2016 [86]
Alberta, Canada	2012–2016	Hs	2 (including metal-tolerance)	Bhatt et al., 2018 [85]
Alberta, Canada	NS	Mh, Pm, HS	Up to 13	Stanford et al., 2020 [18]
Alberta, Canada	2017–2019	Mh, Pm, Hs	Up to 7 *	Conrad et al., 2020 [87]

ARG, antimicrobial resistance gene; Hs, *Histophilus somni*; ICE, integrative and conjugative element; Mh, *Mannheimia haemolytica*; NS, not specified; Pm, *Pasteurella multocida*. * The number of ARGs detected in this study was limited to the PCR assays designed for their detection.

**Table 4 antibiotics-11-00487-t004:** Nucleotide mutations or amino acid changes associated with increased MIC values in *Mycoplasma bovis* isolated from western Canadian beef cattle.

Antimicrobial	Gene	*Escherichia coli* K-12 Numbering	*Mycoplasma bovis* PG45 Numbering	Effect on MIC Values ^1^
Enrofloxacin	*gyrA*	Ser83-Phe	Ser150-Phe	32-fold increase
	*gyrB*	Asp362-Asn	Asp382-Asn	Up to 2-fold increase (slight increase)
	*parC*	Ser80-Ile	Ser91-Ile	2 to 8-fold increase
Tetracyclines	*rrs*	A965T	A956T	Resistance to oxytetracycline. Mutation present in both *rrs* alleles of the 16S rRNA gene: *rrs3* and *rrs4*
		A967T	A958T	
Macrolides	*rrl*	G748A	G788A	Up to 64 and 256-fold increase for TYLT and TIL, respectively. Mutation present in both *rrl* alleles of the 23S rRNA gene: *rrl3* and *rrl4*
		A2059G	A2057G	Up to 32-fold increase for TYLT and TIL. Mutation present in one or both *rrl* alleles of the 23S rRNA gene: *rrl3* and *rrl4*
	*rplV* (protein L22)	Q93H	Q93H	Up to 8 and 16-fold increase for TYLT and TIL, respectively

*Escherichia coli* K-12 substrain MG1655 GenBank accession number: CP014225.1. TIL, tilmicosin; TYLT, tylosin tartrate. ^1^ As per Gautier-Bouchardon, 2018 [50].

**Table 5 antibiotics-11-00487-t005:** Summary of the North American studies that used culture-independent approaches to investigate antimicrobial resistance in the microbiota of the respiratory tract of feedlot cattle.

Geographical Area	Sampling Year (s)	Approach	Authors
Alberta, Canada	2008–2010	qPCR	Holman et al., 2018 [68]
Alberta, Canada	2007–2010	qPCR	Holman et al., 2019 [57]
Alberta, Canada	2016	Metagenomic sequencing	Klima et al., 2019 [69]
Alberta, Canada	2016	qPCR	Guo et al., 2020 [29]

**Table 6 antibiotics-11-00487-t006:** Approaches for the rapid detection of antimicrobial-resistant bacteria.

Technique	Chute-Side Potential	Turnaround Time	Advantages ^1^	Disadvantages	Reference
RPA	Yes	<30 min	Simple; High sensitivity and specificity; Stable when PCR inhibitors present; Fast; Efficient; Offers multiplexing; Cost-effective	Sample processing; Update emerging resistance genes/mutations	[87,99]
LAMP	Yes	30–60 min	Simple; High sensitivity and specificity; Stable when PCR inhibitors present; Fast; Some sample types do not need processing; Cost-effective	High number of primers per target; Limited multiplexing; Update emerging resistance genes/mutations	[100,101]
Metagenomics–Nanopore Minion	Yes	6–8 h	Detection of every pathogen present; Detection of ARG; High sensitivity and specificity; Fast; Cost-effective	Sample processing; Detection of non-pathogenic bacteria genes; Enrichment of low copy number genes by PCR; Host DNA sample contamination	[102]
MALDI-TOF	No	30–180 min	Low-cost consumables; Simple; High sensitivity and specificity; Fast; Various samples in a single run; Cost-effective	Up-front high cost; Requires bacterial isolation or sample processing; Update emerging resistance genes/mutations	[103]

ARG, antimicrobial resistance gene; MALDI-TOF, matrix assisted laser desorption ionization-time of flight; PCR, polymerase chain reaction. ^1^ Bacterial culture or sample processing time not included.

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
