# Peer review of "Bovine Respiratory Disease: Conventional to Culture-Independent Approaches to Studying Antimicrobial Resistance in North America"

_antibiotics, 2022, doi:10.3390/antibiotics11040487_

Round 1

Reviewer 1 Report

This review is a better work to summary the AMR and AMU in the beef industry. Basic on the bovine respiratory disease to dicuss the relationship between increase risk of AMR and possible AMU alternatives. This review is enjoy to read. There are some comments as below:

1: title: suggest add "in North American" after disease

2: L83: single nucleotide polymorphisms (SNPs)

3. in the pert 3 Culture-dependent surveillance studies is too long to read. Suggest authors to integrate to summary them. maybe integrate the part 3 and part 4 as a part will better.

4. The figure 2 is important to this review, but the local site is not clear, please redraft it.

5. The work is better, but it is too long. please short it and easy read.

Author Response

Reviewer 1

1: title: suggest add "in North American" after disease

As suggested, “in North America” has been added to the title (lines 3 and 4).

2: L83: single nucleotide polymorphisms (SNPs)

As pointed out, SNP has been added in brackets (line 84).

  1. in the pert 3 Culture-dependent surveillance studies is too long to read. Suggest authors to integrate to summary them. maybe integrate the part 3 and part 4 as a part will better.

Text in Section 3 is further summarized. Tables 1 and S1 already describe a summary of the North American culture-dependent BRD surveillance studies. For clarity, “discussed in this review (see section 3)” has been added to Table 1 title (line 138) and “culture-dependent” has been added to Table S1 title.

We are reluctant to integrate sections 3 and 4 as they address very different topics,  section 3 describes the bacteria responsible for BRD, while section 4 focuses specifically on the nature of the ICE elements. However, we have reduced the length of section 3 by deleting some of the sentences/paragraphs. Deleted sections are indicated in the marked, returned copy of the manuscript.

  1. The figure 2 is important to this review, but the local site is not clear, please redraft it.

We believe the reviewer is referring to loci site not local site.  Consequently, we have endeavored to improve the resolution of Figure 2 so as to make the loci more clear. We believe this will be clearer when we submit the final jpg images, if the manuscript is accepted for publication.  

  1. The work is better, but it is too long. please short it and easy read.

Thanks for appreciating our work.  We have deleted several sections in the manuscript that were repetitive and that we felt were unnecessary to maintain the impact of the paper.   We have also improved the wording of sentences where we felt it was appropriate. Deleted sections are indicated in the marked, returned copy of the manuscript.

Considering the breadth and complexity of this topic with the involvement of four different bacterial pathogens, resistance phenotype and genomic diversity, a vast array of techniques and experimental designs used in the studies, associated challenges and future prospects we feel that efforts to further condense this review and result in  loss in meaningfulness.  

Reviewer 2 Report

I suggest to improve the paraghraph 3, 4 and 5 adding tables summarizing the main features of cited literature. If available, for each table the European studies included in the study could be highlighted to emphasize any differences  (number of resistant strain and/or ARG detected) between the two geographical regions. In table 4 I suggest to add the references of papers that used the techniques reported.

Author Response

Reviewer 2

I suggest to improve the paragraph 3, 4 and 5 adding tables summarizing the main features of cited literature. If available, for each table the European studies included in the study could be highlighted to emphasize any differences (number of resistant strain and/or ARG detected) between the two geographical regions. In table 4 I suggest to add the references of papers that used the techniques reported.

Our understanding is that with “paragraph 3, 4 and 5” you refer to sections 3, 4 and 5.

Table 1 and Table S1 already include a summary of the studies cited in section 3. Nevertheless, titles of these tables have been modified for further clarity: “discussed in this review (see section 3)” has been added to Table 1 title (line 138) and “culture-dependent” has been added to the title of Table S1.

Following your advice, new summary tables have been added in sections 4 (new Table 3; lines 403, 425-429) and 5 (new Table 5; lines 464, 481-483).

This review is focused on North American studies considering the homogeneity of the animal production system. Discussing AMR in BRD Pasteurellaceae from European studies and how they compared to North American ones is beyond the scope of this comprehensive literature review. European studies are only mentioned in Section 3 with the purpose of highlighting that resistance to macrolides is also high in this geographical region in M. bovis (line 320) and to emphasize B. trehalosi, as a knowledge gap exists about this species in North American studies (line 330). European studies are not stated among culture-dependent surveillance studies of the main Pasteurellaceae bacteria that cause BRD (Section 3.1). Sections 4 and 5 also do not contain citations to European studies.

In Table 4 (now Table 6), references were already included as a footnote in the table. However, we have moved these up and added them as a column within the table for clarity (lines 542 and 545). 

Reviewer 3 Report

The article is comprehensively written and explains the use of antimicrobials and the possible emergence and transmission of AMR among feedlot cattle. Additionally, the authors discussed the surveillance strategies and also suggested the possible strategies to control the spread of AMR or AMR genes like ICE-mediated AMR transfer. 

However, I have only a few questions that need to be discussed; 

  1. Please discuss the possible reasons for "low levels of AMR, but MICs were higher in isolates from dairy-type versus beef-type feedlot cattle" Line 202, page 6
  2. Please also discuss the reasons for "most of the MDR isolates obtained from the USA possessed ICE, which was absent in isolates from Canada". Line 227, page 6.
  3. The authors discussed "MALDI-TOF" in table 4 as culture-independent techniques and similarly in disadvantages highlights the bacterial isolation. This is confusing as MALDI-TOF needs bacterial culturing so how is culture independent? 

Author Response

Reviewer 3

  1. Please discuss the possible reasons for "low levels of AMR, but MICs were higher in isolates from dairy-type versus beef-type feedlot cattle" Line 202, page 6.

As suggested, this point has been discussed: “likely due to higher AMU in dairy as compared to beef calves [1,46]” (lines 219-220).

  1. Please also discuss the reasons for "most of the MDR isolates obtained from the USA possessed ICE, which was absent in isolates from Canada". Line 227, page 6.

As pointed out, this statement has been further discussed: “Higher AMR in isolates from the USA compared to Canada could have been the result of higher AMU due to the larger size and higher animal density of USA feedlots, possibly increasing the incidence of BRD [35].” (lines 246-248).

  1. The authors discussed "MALDI-TOF" in table 4 as culture-independent techniques and similarly in disadvantages highlights the bacterial isolation. This is confusing as MALDI-TOF needs bacterial culturing so how is culture independent?

Thank you so much for pointing out this inconsistency. Text has been modified and added where needed to resolve this confusion and add clarity (see lines 530-541).